

# Technical note: Improving the Initial Conditions of Hydrological Model with Reanalysis Soil Moisture Data

Lingxue Liu [1, †], Tianqi Ao [1,2, †], Li Zhou [2, *]

[1] Institute for Disaster Management and Reconstruction, Sichuan University-Hong Kong Polytechnic University, Chengdu
610065, China
[2] State Key Laboratory of Hydraulics and Mountain River Engineering, College of Water Resource & Hydropower, Sichuan
University, No.24 South Section 1, Yihuan Road, Chengdu 610065, China

† These authors contributed equally to this work.

*Correspondence to*: Li Zhou (zhouli.scu@gmail.com)

**Abstract.** The initial conditions (e.g., soil moisture content) of the hydrological model, which is usually obtained from the
warm-up of the hydrological modeling, significantly impact the simulation efficiency. However, spending the valuable data
in warm-up instead of calibration and validation is luxurious. In order to improve hydrological simulation efficiency in the
case of no warm-up phase, this paper proposes a methodology to fill the gap via improving the initial conditions of the
hydrological model using an alternative global soil moisture dataset. Specifically, three soil moisture (SM) variables of the
initial conditions from the Block-wise use of the TOPMODEL (BTOP) model and ERA5-Land reanalysis data were adopted
and conducted correlation analysis. Several traditional curve-fitting functions and the state-of-art technical, long-short term
memory (LSTM), were applied to develop the relationship between BTOP and ERA5-Land SM variables in the Fuji and
Shinano River Basin, Japan. Furthermore, four configured hydrological simulations evaluated the benefits of the proposed
methodology for improving the initial conditions. As a result, LSTM outperforms the traditional curve-fitting method in
constructing the relationship between variables in time and space. Moreover, the hydrological simulation cases using the
initial conditions related to the SM from the ERA5-land performs better than the case without the warm-up phase, and the
simulated discharge process approaches the "optimal" case with the warm-up phase. It is confirmed that the proposed
methodology helps improve the initial conditions of the hydrological model using reanalysis soil moisture data.

**Keywords.** *BTOP model, ERA5-Land, model warm-up, Long-short term memory (LSTM), ungauged basin*

## 1 Introduction

Hydrological model is an essential tool to explore the physical law of hydrological process (Refsgaard, 1997; Senarath et al.,
2000) and to provide valuable simulated results for various purposes such as drought and flood monitoring (Chen et al.,
2018), water resources and irrigation management (Grové, 2019), and water environmental pollutant migration (Basheer,
2018). It needs to be calibrated to minimize the uncertainty before application (Gupta et al., 2009). One of the critical



impacts on hydrological modeling is the initial conditions (e.g., soil moisture content) which affect the simulation efficiency (e.g., stability and convergence) significantly, especially at the beginning of the hydrological simulation (Berthet et al., 2009; Bui et al., 2021). However, the issue of initial conditions has not been well solved due to various uncertainties from complex natural conditions, hydrological processes, and insufficient data (Beven and Binley, 1992; Cloke et al., 2003; Beven, 2006).

Generally, the initial conditions are often acquired from the warm-up of the hydrological model, which is a process adjusting the initial conditions of the model from the estimated state to the "optimal" state to reduce the impact of the initial state on the hydrological simulation (Kim et al., 2018). Usually, it is challenging to balance the period of model warm-up between data utilization and warm-up efficiency. Furthermore, the model warm-up period for daily simulation was usually set to one or two years at least (Boufala et al., 2019; Paul et al., 2020; Yu et al., 2019), or even several years (Erraioui et al., 2020;

Carlos Mendoza et al., 2021). This is highly extravagant to the vast ungauged basins worldwide, especially for the developing countries and mountainous areas. Therefore, it would significantly improve the hydrological simulation if the warm-up period could be shortened or skipped.

Soil moisture (SM) is a crucial variable among the initial conditions which affects the hydrological processes significantly as essential as precipitation (Kim et al., 2018; Niroula et al., 2018). Compared with other water cycle components, although the

total mass of soil water content is small, it affects the climate system by controlling the interaction of water, energy, and carbon flux between the land surface and the atmosphere (Seneviratne et al., 2010). Although the International Soil Moisture Network (ISMN) established a global in-situ soil measurement database for the use of improving satellite SM products and climate, land surface, and hydrological models (Dorigo et al., 2011, 2013), it has apparent limitations due to the poor spatial coverage and uneven distribution worldwide (Miralles et al., 2012; Nguyen et al., 2017). With the advancement of

technology, the satellite SM products and reanalysis data from the land surface model (LSM) have been significantly improved (Mousa and Shu, 2020; Wu et al., 2021; Dorigo et al., 2017; Muñoz Sabater et al., 2021), which allows the possibility to connect them with hydrological model SM variables.

Considerable work has been done about the initial conditions of the hydrological model on some subjects, such as the impact on ensemble streamflow prediction (Li et al., 2009; Roulin and Vannitsem, 2015; Piazzi et al., 2021; Donegan et al., 2021),

using data assimilation to improve the initial conditions for ensemble streamflow prediction (Dechant and Moradkhani, 2011; Cho and Kim, 2022; Muñoz et al., 2022), and employing alternative SM data (e.g., satellite, reanalysis) into the hydrological model directly (Muñoz et al., 2022; Massari et al., 2014; Setti et al., 2020). To our best knowledge, however, none of them addressed the issue of how to improve the initial conditions (e.g., soil moisture content) of the hydrological model with other datasets via developing their relationship. Therefore, this paper aims to propose a methodology to fill this gap, which could

improve the hydrological simulation by obtaining the initial conditions of the hydrological model from other datasets with proper process. In Sect. 2, we describe the material used in this study: two river basins in Japan, the Block-wise use of the TOPMODEL (BTOP) model, and the ERA5-Land dataset. The methodology is provided in Sect. 3, which employs six curve-fitting functions and the state-of-art deep learning technology, long short-term memory (LSTM) method to develop the



relationship between BTOP and ERA5-Land SM variables, and a configuration of hydrological simulation is designed for
illustrating the importance of initial conditions, model warm-up, and the benefits of the proposed methodology. Sect. 4
describes the results and discussion of correlation analysis, relationship development, hydrological simulation, etc. Finally,
we present the conclusions in Sect. 5.

## 2 Material

### 2.1 Study area

In this paper, we employed two major river basins in Japan as they have adequate hydro-meteorology data to achieve the
goals of this study. As shown in Fig. 1, the Fuji River Basin (FRB) and Shinano River Basin (SRB) are located in the central
part of the Honshu island. They originate from the Japanese Alps, known as Japan's peak area.

The FRB flows through Nagano, Yamanashi, and Shizuoka prefectures, with a river length of 128 km and a drainage area of
approximately 3570 km$^2$. It flows into the Pacific Ocean at Suruga bay, and the downstream section Kitamatsuno has a mean
annual flow of 63.2 m$^3$/s. The average temperature of summer and winter are 26℃ and 3℃, respectively (Shrestha and
Kazama, 2006). Kofu Plain, which lies upstream of FRB, receives an average annual rainfall of 1100 mm. The middle and
lower reaches of FRB have an average annual rainfall of 2000 and 2500 mm, respectively. The basin terrain is steep, with
90% of the area being mountainous or hilly. There are many peaks in the basin, including Japan's highest and world-famous
mountain, Mt. Fuji, with an altitude of 3776 m.

The Shinano River is Japan's longest (367 km) and third-largest drainage area basin (11900 km$^2$). It originates at the foot of
Kobushi Mountain in the Alps of Honshu Island, flows through Nagano and Niigata prefectures, and enters the Sea of Japan.
The annual average flow of the Ojiya section is 503 m$^3$/s. The upper part of the SRB is called the Chikuma River Basin
(CRB), which has a river length of 214 km and a drainage area of 7163 km$^2$, accounting for 58% and 60% of the SRB,
respectively. In the upstream of CRB, it is surrounded by mountains, and there is only 10% of the land is flat for agriculture.
The inland climate is remarkable, and the precipitation is low. The annual average precipitation in Nagano City is only 938
mm. However, the downstream part of SRB on the Niigata side has a unique climate in coastal areas of Japan. The annual
average precipitation in Nagaoka City is 2310 mm. The abundant water and fertile soil make this area one of the best rice-
producing areas in Japan.

### 2.2 Hydrological model: BTOP model

#### 2.2.1 Brief introduction

The Block-wise use of the TOPMODEL (BTOP model) is based on the well-known semi-distributed hydrological model-
TOPMODEL (Beven and Kirkby, 1979). It has been continually developed (Ao et al., 2006; Zhou et al., 2006; Takeuchi et
al., 2008; Zhang et al., 2018) and applied to various basins worldwide (Magome et al., 2015; Gusyev et al., 2016; Liu et al.,



2020) for varies of purposes such as water resource management (Hapuarachchi et al., 2008), flood and drought monitoring
(Zhou et al., 2021) since 1999 when it was first proposed in the University of Yamanashi, Japan (Takeuchi et al., 1999; Ao et al., 1999). The BTOP model is a semi-physically, reliable, and straightforward hydrological model, and its parameters have physically interpretation which could represent the influence of underlying surfaces such as vegetation, land use, soil properties, and soil moisture (Takeuchi et al., 2008; Wang et al., 2010; Zhu et al., 2021). These features allow the possibility of connecting the model simulated SM variables with other datasets such as reanalysis and satellite SM data.

## 100 2.2.2 Initial conditions and soil moisture-related variables

As shown in Fig. 2a, the BTOP model defines the area above the ground surface covered by the plant canopy as a vegetation area and divides the subsurface aeration zone into three parts: the root zone, the unsaturated zone, and the saturated groundwater zone (Takeuchi et al., 2008). Table 1 describes the initial condition variables of the BTOP model, which could represent the basic information of underlying surface such as discharge, soil moisture, snow cover at each grid. This study
focuses on the soil moisture related variables in the BTOP model: root zone storage ($S_{rz}$, m), unsaturated zone storage ($S_{uz}$, m), and saturation deficit ($SD$, m).

① Storage in root zone: $S_{rz}$

The root zone storage at grid $i$ and time step $t$ is calculated by the equation below:

$$S_{rz}(i,t)=S_{rz}(i,t-1)+\Delta t\left(P_a(i,t)-q_{ofh}(i,t)-ET(i,t)-q_{rz}(i,t)\right) \tag{1}$$

where $S_{rz}(i,t-1)$ is the root zone storage at time step $t-1$; $P_a(i,t)$ is the net rainfall; $q_{ofh}(i,t)$ is the hortonian overland flow; $ET(i,t)$
is the actual evapotranspiration from root zone; $q_{rz}(i,t)$ is the storage excess of the root zone at time step $t$.

② Storage in unsaturated zone: $S_{uz}$

The unsaturated zone storage at grid $i$ and time step $t$ can be represented by the following equation:

$$S_{uz}(i,t) = S_{uz}(i,t-1)+\left[q_{rz}(i,t)-q_{of}(i,t)-q_v(i,t)\right]\Delta t \tag{2}$$

where $S_{uz}(i,t-1)$ is the unsaturated zone storage at time step $t-1$; $q_{of}(i,t)$ is the saturation excess runoff flux; $q_v(i,t)$ is groundwater recharge at time step $t$.

③ Saturation deficit: $SD$

The saturation deficit at grid $i$ and time step $t$ is updated by the following equation:

$$SD(i,t) = SD(i,t-1)-\left(q_{rz}(i,\mathrm{t})+q_v(i,t)-q_b(i,t)\right)\Delta t \tag{3}$$

where $SD(i,t-1)$ is the saturation deficit at time step $t-1$; $q_b(i,t)$ is the base flow at time step $t$.



### 2.3 Model input data

#### 2.3.1 Precipitation and discharge data

This study collected daily precipitation and discharge data from 2002 to 2011 from the Japan Meteorological Agency (JMA) and the Ministry of Land, Infrastructure, Transport and Tourism (MLIT). As shown in Fig. 1, 26 and 77 precipitation stations were selected in FRB and SRB, respectively. Moreover, we employed two and seven discharge stations in FRB and SRB according to the basin area and data quality, making two and seven subbasins for each. For the model simulation, we set 2002 as the model warm-up period. In addition, 2003-2007 and 2008-2011 are set as calibration and validation periods.

#### 2.3.2 Other input data

We adopted 500 m and 1000 m resolution in FRB and SRB, respectively, for the model simulation, considering the data representation and computing time. Therefore, all the following data were resampled to the two resolutions above. The Digital Elevation Model (DEM) was obtained from Multi-Error-Removed Improved-Terrain (MERIT) DEM with an original resolution of three seconds (90 m at the equator) (Yamazaki et al., 2017). The BTOP model employed the MODIS-IGBP

Land Cover map (original resolution: 500 m) as land cover data (Friedl and Sulla-Menashe, 2019). Normalized Difference Vegetation Index (NDVI) and Leaf Area Index (LAI) came from the National Oceanic and Atmospheric Administration (NOAA) of the United States (Vermote et al., 2014) with a resolution of 0.05 degrees. Soil properties were obtained using the soil map (at a scale of 1:5 million) of the Food and Agriculture Organization (FAO). The Climate Research Unit (CRU) provided meteorological data (Harris et al., 2020) with a resolution of 0.25 degrees, such as temperature, radiation, humidity,

wind speed, and vapor pressure, for the evaporation module of BTOP model (Zhou et al., 2006) to generate potential interception evaporation ($PET_0$) and potential evapotranspiration (PET).

### 2.4 Reanalysis soil moisture data: ERA5-Land

The European Centre for Medium-Range Weather Forecasts (ECMWF) produces an enhanced global dataset for the land component of the 5[th] generation of European ReAnalysis (ERA5), called ERA5-Land (Muñoz Sabater et al., 2021). It covers

the same period (January 1950 to near real-time) and temporal resolution (hourly) as ERA5 (Hersbach et al., 2020). Compared with ERA5, ERA5-Land runs at enhanced resolution (0.1°, 9 km vs. 31 km in ERA5) without coupling to the ECMWF's Integrated Forecasting System, making it computationally affordable and lighter to handle. Moreover, it better describes the hydrological cycle, particularly with enhanced soil moisture, allowing it to broadly utilize various purposes such as SM monitoring and enhancing hydrological simulation. Unfortunately, to the authors' best knowledge, the research

related to the SM from the ERA5-Land has not been reported in Japan. However, in some areas which have similar climatic characteristics with the study area in this paper, the ERA5-Land SM data showed a better performance than many other datasets such as Global Land Data Assimilation System (GLDAS-2.1) (Wu et al., 2021), Advanced Scatterometer (ASCAT)





 and Soil Moisture and Ocean Salinity (SMOS) (Pablos et al., 2021). Therefore, the ERA5-Land dataset is worthy of being used to fulfill the objectives of this study.

Figure 2b shows the SM layer structure of ERA5-Land. It is divided into four layers: 0-7cm, 7-28cm, 28-100cm, and 100-289 cm. Moreover, each layer contains the soil moisture content (water storage) of *S1* to *S4*. We downloaded the hourly SM data from 2002 to 2011 from the Climate Data Store, Copernicus program (Muñoz Sabater, 2019). Then we shifted the ERA5-Land SM data to the Japan Standard Time (JST, UTC+9) and converted hourly to daily data to consistent the temporal with BTOP simulated variables. Moreover, to compare ERA5-Land SM with BTOP SM variables, we converted

the original unit ($m^3/m^3$) to meter by multiplying the depth of each corresponding layer under the assumption that the water content is evenly distributed in each layer (Brouwer et al., 1985).

## 3 Methodology

We propose a methodology shown in Fig. 3 to achieve the objectives of this study. Firstly, four hydrological simulation cases are configured to build a comprehensive experiment and evaluation system for proving the importance of model warm-

up for hydrological simulation (Case 1 and 2), and whether it is possible to utilize alternative SM data (ERA5-Land) to improve the initial conditions of the hydrological model or not (inter-comparison of four cases). As shown on the right side of the framework, the SM variables of BTOP model and ERA5-Land are comprehensively analyzed in temporal and spatial. Then six traditional curve-fitting functions and cutting-edge technology, long-short term memory (LSTM), are used for developing the relationship of SM variables between BTOP model and ERA5-Land at both basin- and grid-scale. Finally, a

comprehensive evaluation is conducted for verifying the relationship development of SM variables between BTOP model and ERA5-Land, and an inter-comparison and evaluation is carried out for hydrological simulations with different initial conditions and warm-up processes.

### 3.1 Case configuration of hydrological simulations

We configure four hydrological simulation cases for FRB and SRB. The details are shown in Table 2. They share the exact

calibration (2003-2007) and validation period (2008-2011), and all cases are auto-calibrated by shuffled complex evolution (SCE-UA) method (Duan et al., 1994) with approximately ten thousand irritations for eight simulations each (four cases for two basins). Case 1 employs 2002 as the warm-up period. We consider its simulated variables are the most representative of the hydrological model. Therefore, it is regarded as the "optimal" case and provides the referee SM variables for the correlation analysis and relationship development with ERA5-Land SM data. Case 2 is the control test conducting the

simulation without warm-up to verify the warm-up effect for the hydrological model. Case 3 and 4 take the SM variables processed from processed ERA5-Land by using traditional curve-fitting and LSTM methods as the initial condition of BTOP model, respectively.





## 3.2 Correlation analysis

Before the relationship development of BTOP and ERA5-Land SM variables, a comprehensive correlation analysis should
be carried out at different spatial (grid, sub-basin, basin) and temporal (daily and annual average daily) scales, which
employs Pearson correlation coefficient ($R$) as the performance index. The daily SM data is from 2003-2011, covering the
calibration and validation period. It should be noted that the outputs of BTOP model (500 and 1000 m for FRB and SRB,
respectively) are resampled to 0.1° to be consistent with ERA5-Land, and all variables are processed using Min-max
normalization technical (Jain et al., 2005; Antanasijevic et al., 2014). Three correlation analysis experiments (EXP) are
conducted as the following description.

(1) EXP 1: analyzing one by one.

Three BTOP SM variables ($S_{rz}$, $S_{uz}$, $SD$) are analyzed with four ERA5-Land variables ($S1$, $S2$, $S3$, $S4$) successively.

(2) EXP 2: relating BTOP SM variables to the combination of ERA5-Land SM variables.

Many papers regarded the part between the ground surface to 100 cm below as the root zone (Bai et al., 2021; Pradhan, 2019;
Qi et al., 2019). Moreover, in the BTOP model, $S_{rz}$ represents the storage in the root zone. Therefore, from the physical
concept and the water content structure shown in Fig. 2, it is worth connecting $S_{rz}$ with the sum of $S1$, $S2$, and $S3$, denoted as
$Sa$ in this paper. On the other hand, $SD$ represents the saturation deficit in the BTOP model. Thus, we assume that its concept
is similar to the value of ERA5-Land soil depth (289 cm) minus $Sa$, which is expressed as $Sb$.

(3) EXP 3: relating $S_{uz}$ to $S_{rz}$ and $SD$.

As for the $S_{uz}$, there is no apparent physical meaning to support its connection to ERA5-Land SM variables. Nevertheless,
suppose we could get the relationship between one of the BTOP variables and ERA5-Land SM variables. In that case, it is
not challenging to develop a relationship among the BTOP SM variables as they usually have a strong connection. Therefore,
experiment three is designed to connect the $S_{uz}$ with $S_{rz}$ or $SD$. This could also be an alternative solution for other
hydrological models when conducting this methodology.

## 3.3 Relationship development methods

This study employs two methods (curve-fitting and LSTM) to develop the relationship of SM variables between BTOP and
ERA5-Land at two spatial scales: grid- (0.1°) and basin-scale. Specifically, the grid-scale applies the relationships developed
by each grid to the corresponding grid, while the basin-scale uses the relationship developed by basin-average data for each
grid. We take the model calibration period (2003-2007) and validation period (2008-2011) as the training and test period for
developing the relationship, respectively.

### 3.3.1 Curve-fitting functions

Curve-fitting is a process of fitting the measured points by the appropriate functions to minimize the distance between the
observed and fitted points (Ueng et al., 2007; Adnan et al., 2020). The commonly used curve types such as polynomial,



logarithmic, and power functions are always hard to express complex data distributions well (Pourkarimi et al., 2011). The spline fitting was proposed based on the practical piecewise polynomials (Dierckx, 1981). The widely applied and improved cubic spline fits each piece segmented by the knots with cubic polynomial, which is similar to the piecewise polynomials. However, there are some restrictions that the polynomials, their first-order and second-order derivatives are all continuous at the knots to generate a smooth curve (Lavery, 2002, 2000; Zhanlav and Mijiddorj, 2018). Moreover, due to the unstable polynomial functions and the fewer measured points, over-fitting often occurs in the boundary region. Sequentially, an additional restriction that the function outside the boundary knots is linear was added, and the corresponding spline is called a natural spline. It allows the polynomials to extend smoothly beyond the boundary knots (Huang et al., 2018).

Combined with the scatter distribution of the well-correlated variable combinations obtained by the correlation analysis (as shown in Sect. 4.2), we selected six commonly used functions in Table 3. They are applied for the relationship development at grid- and basin-scale in Sect. 4.3.

### 3.3.2 Long short-term memory (LSTM)

LSTM is developed to address the problem of vanishing gradient in recurrent neural networks (RNN), and has been widely used in various kinds of tasks, including speech recognition and sentence embedding (Arslan and Barışçı, 2019; Palangi et al., 2016; Graves et al., 2013), correlation analysis (Deng et al., 2020; Yang et al., 2020), and hydrometeorological forecast (Yin et al., 2021; Ni et al., 2020). LSTM has a special internal structure design which includes two states (cell state, hidden state) for information storage and three gates (input gate, forget gate, and output gate) for information addition or deletion, making it a strong learning ability and applicable for sequence data learning (Yu et al., 2019; Sherstinsky, 2020). Referring to Keras (Chollet and Others, 2015), a deep learning algorithm written in python, LSTM conducted in this study is described in Fig. 4. The input variables include precipitation (P), potential evapotranspiration (PET), potential intercept evaporation (PET$_0$), leaf area index (LAI), and the water storage of four layers ($S1$, $S2$, $S3$, $S4$) from the ERA5-Land, while the outputs are $S_{rz}$, $S_{uz}$ and $SD$ of BTOP model.

## 3.4 Evaluation scheme

### 3.4.1 Evaluation of the fitting method and developed relationship

(1) General evaluation criteria

The Pearson correlation coefficient ($R$) is commonly used for correlation analysis as it can well represent their relationship strength between two variables (Al-Yaari et al., 2017; Gruber et al., 2020). Moreover, several indicators are selected as evaluation criteria, such as relative mean absolute error ($rMAE$), relative root mean square error ($rRMSE$), normalized standard deviation ($NSD$), normalized root mean square deviation ($NRMSD$), and coefficient of determination ($R^2$). The following equations show the details of the evaluation indicators:



$$R = \frac{\sum_{i=1}^{n}\left(X_i^B - \overline{X^B}\right)\left(X_i^E - \overline{X^E}\right)}{\sqrt{\sum_{i=1}^{n}\left(X_i^B - \overline{X^B}\right)^2}\sqrt{\sum_{i=1}^{n}\left(X_i^E - \overline{X^E}\right)^2}} \tag{4}$$

$$rMAE = \frac{MAE}{\overline{X^B}} = \frac{\sum_{i=1}^{n}\left|X_i^E - X_i^B\right|}{\sum_{i=1}^{n}X_i^B} \tag{5}$$

$$rRMSE = \frac{RMSE}{\overline{X^B}} = \sqrt{\frac{n\sum_{i=1}^{n}\left(X_i^E - X_i^B\right)^2}{\left(\sum_{i=1}^{n}X_i^B\right)^2}} \tag{6}$$

$$NSD = \sqrt{\frac{\sum_{i=1}^{n}\left(X_i^E - \overline{X^E}\right)^2}{\sum_{i=1}^{n}\left(X_i^B - \overline{X^B}\right)^2}} \tag{7}$$

$$NRMSD = \frac{\sum_{i=1}^{n}\left|X_i^s - X_i^o\right|^2}{\sum_{i=1}^{n}X_i^{o\,2}} \tag{8}$$

where $X^B$ are the BTOP SM variables; $X^E$ are the ERA5-Land SM variables; $i$ is the time step.

(2) Selection scheme of curve-fitting functions

To uniform the performance of each curve-fitting function, we develop an evaluation scheme as described as follows: Firstly, normalizing the five indicators shown above using Min-max normalization technical (Jain et al., 2005; Antanasijevic et al., 2014) to get the score values between 0 and 1. Secondly, it gives the exact weight of 0.2 to five each index to keep the optimal score as one still. Thirdly, assigning weights of 0.7 and 0.3 to test and training period, respectively, since we value

the test period more. To this point, the calculated scores of each function have been completed with an optimal value of 1. It should be noted that the fitting process is only conducted at the basin scale in the selection phase.

### 3.4.2 Evaluation of the hydrological simulation

As for the evaluation of hydrological simulation, not only the Nash-Sutcliffe Efficiency (*NSE*) (Nash and Sutcliffe, 1970) but also the improved Kling–Gupta efficiency (*KGE′*) (Gupta et al., 2009) are employed as the criteria to evaluate the

hydrological simulation efficiency among different configured cases. The equations are shown below:

$$NSE = 1 - \frac{\sum_{i=1}^{n}\left(Q_i^S - Q_i^O\right)^2}{\sum_{i=1}^{n}\left(Q_i^O - \overline{Q}_i^O\right)^2} \tag{9}$$



$$KGE' = 1 - \sqrt{(r-1)^2 + (\beta - 1)^2 + (\gamma - 1)^2} \tag{10}$$

where $Q^s$ and $Q^o$ represent the simulated and observed discharge respectively; $r$ is the correlation coefficient, $\beta$ is the bias ratio, $\gamma$ is the variability ratio.

## 4 Results and discussion

### 4.1 SM variables of BTOP and ERA5-Land

As we described the methodology in Fig. 3, BTOP SM variables ($S_{rz}$, $S_{uz}$, $SD$) acquired from hydrological simulation Case 1 are regarded as the referee data of model SM variables. This section analyzed them and four ERA5-Land SM variables ($S1$, $S2$, $S3$, and $S4$) from 2003 to 2011 at both temporal and spatial scales. According to the hydrological simulation performance of each section (see details Table S1), however, we only adopted the results of two stations in FRB (subbasin: FRB-1, FRB-2) and four stations in SRB (subbasin: SRB-1, SRB-2, SRB-3 SRB-4) for the following analysis process and hydrological
simulations due to the poor simulated performance in SRB-5, SRB-6, and SRB-7.

As shown in Fig. 5 and Fig. 6, the SM variables of BTOP and ERA5-Land have a certain relationship, regardless of the temporal or spatial scale. Specifically, despite the absolute values, in the view of temporal scale (Fig. 5), daily changes of BTOP are more dramatic than ERA5-Land, and the ERA5-Land have a better interannual variability than BTOP in terms of annual average daily scale. From the aspect of spatial distribution (Fig. 6), the BTOP variables have a more evident and
specific distribution than ERA5-Land due to a higher resolution. According to the definition of $SD$ in BTOP (Takeuchi et al., 2008), the values of $SD$ are negative in some areas (central FRB and SRB) due to land use of city area.

It should be noted that, since the BTOP and ERA5-Land variables are come from hydrological and land surface models, and the models' fundament are based on many conception assumptions instead of actual physical law (Liang et al., 1994; Albergel et al., 2012; Frassl et al., 2018; Muñoz Sabater et al., 2021), there is no "truth" value in this study. Moreover, the
absolute values of these seven variables are different; therefore, it is necessary to conduct several experiments of correlation analysis in the following section. In addition, as the spatial characteristics of BTOP and ERA5-Land are different, it is also necessary to develop their relationship at grid-scale instead of basin-scale only.

### 4.2 Correlation analysis of SM variables

#### 4.2.1 EXP 1: Correlation of $S_{rz}$, $S_{uz}$, $SD$ and $S1$, $S2$, $S3$, $S4$

Experiment one analyzed the correlation between BTOP and ERA5-Land SM variables successively to understand the relationships among them better. Figure 7 shows the scatterplots of the correlation results of EXP 1 at basin scale, while Fig. S1 and S2 are at sub-basin scale, which is similar to the basin scale. The $S_{rz}$ has a significant positive correlation with $S1$, $S2$, and $S3$ in different (sub-) basins at a daily scale (the red dots and lines in Fig. 7**a** and **b**, the values of $R$ are around 0.5). In contrast, $SD$ negatively correlates with ERA5-Land variables (the blue dots and lines in Fig. 7**a** and **b**), especially the




correlation coefficient with *S4* where the absolute value reaches 0.7 in FRB at daily scale. As for the $S_{uz}$, there is no apparent
      correlation with ERA5-Land at daily scale. Given the annual average daily scale (Fig. 7**c** and **d**), the correlation between $S_{uz}$
      and ERA5-Land is stronger than the daily scale. However, it is not enough to support the relationship development between
      these two variables. Therefore, we conducted EXP 3 for $S_{uz}$ which has poor connections with ERA5-Land SM variables, and
      the results are shown in Sect. 4.2.3.

**4.2.2 EXP 2: Correlation of $S_{rz}$ and *Sa*, *SD* and *Sb***

      Although $S_{rz}$ and *SD* have good correlations with each layer of ERA5-Land SM variables, the relationship development
      should use more reasonable correlated variable combinations with reliable physical meaning or interpretation as much as
      possible. Thus, EXP 2 illustrated the correlations of *Srz* and *Sa*, *SD* and *Sb,* which were constructed based on the physical
      interpretations. Figure 8 and Figure S3 show the basin- and subbasin-scale results, respectively. Same as EXP 1, the
performance of the subbasin scale is similar to the basin scale. Although the performance of the annual average daily scale
      (Fig. 8b) is not satisfied, the results of daily scale (Fig. 8a) show a considerable close correlation as the values of *R* are more
      stable, and the dots are more gathered than EXP 1 (Fig. 7a and b). It indicates that EXP 2 is more appropriate for the
      relationship development of $S_{rz}$ and *SD* than EXP 1.

      **4.2.3 EXP 3: Correlation of $S_{uz}$ and $S_{rz}$, *SD***

As the results of EXP 1 that no apparent connections are shown between $S_{uz}$ and the ERA5-Land SM variable, EXP 3 was
      conducted to explore the relationship among BTOP variables since they have strong connections due to the model structure
      (Takeuchi et al., 2008; Hapuarachchi et al., 2008). The daily and annual average daily scatterplots of EXP 3 at basin scale
      are shown in Fig. 9, while Fig. S4 shows the results of the subbasin scale, which are basically consistent with the basin scale.
      In the view of daily scale, as shown in Fig. 9a, no apparent relationships are demonstrated between $S_{uz}$ and $S_{rz}$ or *SD*.
Nevertheless, from the aspect of the annual average daily scale, Fig. 9b shows a strong connection between $S_{uz}$ and *SD*, with
      the absolute correlation coefficients of 0.89 and 0.56 in FRB and SRB, respectively. Therefore, it is reasonable to take *SD* to
      develop the relationship with $S_{uz}$ using the correlation information at the annual average daily scale.

      It should be noted that the poor performance of simulated $S_{uz}$ in this study might be caused by the model uncertainties or
      some other unknown reasons at the current stage. We must fix the possible problem in the BTOP model in future work.
However, EXP 3 still shows an alternative way to establish the relationship development for the variables that do not have
      adequate connections with other datasets like ERA5-Land.

      **4.2.4 Performance of correlation coefficients at grid-scale**

      Figure 10 shows the boxplots of the absolute correlation coefficients of all SM variables combinations from EXP 1-3 at grid-
      scale. In the view of $S_{rz}$ as shown in Fig. 10a, the daily scale performs better than the annual average daily scale, and the
results of *Srz-S1*, *Srz-S2*, *Srz-S3* are similar with $S_{rz}$ -*Sa* at daily scale, which has a median *R* round 0.5. However, *Srz-Sa* still





shows a slight advantage over others as it has physical interpretations explained in EXP 2, Sect. 3.2. Figure 10b shows the correlation results of *SD*. It's clear that *SD-Sb* outperforms others at daily scale. In the view of the annual average daily scale, it has a wide range of *R* and lower median value than daily scale as some grids do not have a good relationship with corresponding SM variables. As for $S_{uz}$, which does not have a considerable correlation with ERA5-Land SM variables, we

conducted an additional experiment to connect the BTOP variables shown in Fig. 10c together with all correlation results of $S_{uz}$. The correlation shows a significant improvement while using $S_{uz}$-*SD* at the annual average daily scale, in which the highest absolute *R* reaches 0.92 in the case of FRB-1.

Moreover, Fig. 11 shows the spatial distribution of the absolute correlation coefficients at grid-scale in FRB and SRB. From the aspect of $S_{rz}$ and *SD*, their relationships with *Sa* and *Sb* are prior choices for the following relationship construction under

consideration of performance and physical mechanism. Looking at the results of $S_{uz}$-*SD*, there are some areas with pretty high correlations, while the city, plain areas with bad results. However, considering the poor connections between $S_{uz}$ and ERA5-Land, $S_{uz}$-*SD* is still the best choice at the current stage. In summary, according to the correlation analysis results, the relationships of $S_{rz}$ and *Sa*, *SD* and *Sb*, and $S_{uz}$ and *SD* are chosen to develop the relationship formulas.

**4.3 Relationship development of SM variables between BTOP and ERA5-Land**

This section shows the results of the selection of six curve-fitting functions, the developed relationships using the selected curve-fitting functions and LSTM, and their performance at grid- and basin-scale.

**4.3.1 Selection of curve-fitting functions**

Table 4 shows the scores of each curve-fitting function in the two basins. In addition, Table S2-S4 present the developed formulas of three BTOP SM variables with different curve-fitting functions. For $S_{rz}$-*Sa*, the natural cubic spline outperforms

other functions. In the view of *SD-Sb*, the quadratic polynomial has the best scores. They have a strong linear relationship, making the polynomial functions fit the relationship development best. From the $S_{uz}$-*SD* aspect, first-order polynomial and quadratic polynomial perform best in FRB and SRB, respectively. It is reasonable that the blue dots in Fig. 9b show a more robust liner relationship in FRB than it is in SRB. Therefore, we employed the optimal curve-fitting functions for each basin and BTOP SM variables, as shown in the bold number in Table 4.

**4.3.2 Performance evaluation of curve-fitting and LSTM**

Figure 12 shows the performance evaluation results of selected curve-fitting and LSTM in FRB and SRB using the Talyor diagram (Taylor, 2001). Generally, the LSTM with grid-scale (blue dot) is the best among these relationship development methods, and the results developed by grid-scale outperform those by basin-scale in the test period as the dense circles are closer to REF than cross markers. Although, the LSTM at basin-scale (purple cross) shows a slighter poor performance than

grid-scale (purple dot) in the test period. The test period achieves satisfying results with basin-scale (blue circle). Moreover, the LSTM with basin-scale also shows good spatial performance in Fig. S5**,** which presents a certain day of the spatial




distribution of BTOP SM variables and their predicted ones by curve-fitting and LSTM at both grid- and basin-scale. It demonstrates that the LSTM has the ability to represent spatial characteristics even conducting at basin scale. One reason is that the input factors include the variables that have spatial information, such as evapotranspiration and leaf area index. Thus,

when this methodology is applied to a large basin or area, it is recommended to use LSTM with basin-scale to reduce the computation. From the aspect of $S_{uz}$ shown in Fig. 12c, it is evident that both methods show poorer performance than $S_{rz}$ and $SD$. As we mentioned in Sect. 4.2.3, the performance of $S_{uz}$ in the BTOP model needs to be improved in future work. Accordingly, the fitting results from the developed relationships based on LSTM and the curve-fitting method at grid-scale are applied to the configured hydrological simulations.

## 4.4 Inter-comparison and evaluation of configured hydrological simulations


Figure 13 shows the evaluation results of four configured hydrological simulations described in Table 2, and the specific values are presented in Table S5. Although it has a few differences in the view of $R$ in calibration and validation period shown in the third column of Fig. 13, Case 2 (blue lines) simulated without warm-up phase performs the worst from the aspect of NSE and KGE' compared to the referee Case 1 (red lines) shown in the first two columns in Fig. 13. It indicates

that the warm-up process is significantly necessary for the hydrological simulations. In the calibration and validation period shown in Fig. 13a and b, Case 3 and 4 considerably improved the efficiency of runoff simulations compared to Case 2 except SRB-4, and the LSTM performs slightly better than the curve-fitting method.

To further address the impact of the initial conditions on the hydrological simulation, Fig. 13c shows the results of the year 2003, which is the first year in the calibration period after the warm-up period. Case 3 and 4 utilizing ERA5-Land SM data

to obtain the initial conditions significantly improve hydrological simulation efficiency. Specifically, given the mean value of seven simulated sub-basins shown in Table S5, Case 2 is the worst with a mean $R$ of 0.45, while Case 1 is the best one with 0.71. Case 3 and 4 with $R$ values of 0.67 is similar to Case 1. It indicates that the hydrological simulation could be considerably improved with improving initial conditions, especially in the warm-up period.

In general, the hydrological simulation results of BTOP model are considerably improved with improving initial conditions

acquired from ERA5-Land SM variables compared to Case 2 which does not have a warm-up phase. This study conducted the optimal curve-fitting function and LSTM method to determine the relationship between BTOP and the ERA5-Land SM variable. Currently, lots of references have addressed the close correlation among satellite, reanalysis, model-simulated, or in-situ SM variables (Beck et al., 2020; Zhang et al., 2021; Ling et al., 2021). Therefore, the authors believe this methodology could be applied to other models or alternative datasets. Nonetheless, more work needs to be done in the future

to address more models and datasets as a single model has obvious limitations (Orth et al., 2015).



## 5 Conclusions

In this paper, we proposed a methodology that well-utilizes the alternative global soil moisture data to improve hydrological simulation efficiency without warm-up by providing the initial conditions of the hydrological model with proper processes. The BTOP model and ERA5-Land reanalysis data were selected to represent the hydrological model and global soil moisture

dataset. Six discharge stations divided the subbasins and evaluated hydrological simulation in the Fuji River Basin and Shinano River Basin, Japan. Then, we comprehensively analyzed the correlation of BTOP and ERA5-Land SM variables and developed their relationships using traditional curve-fitting functions and the LSTM method. Finally, we demonstrate the benefits of the proposed methodology on the hydrological simulation. The conclusions are as follows:

(1) The warm-up is necessary for hydrological simulation if there is no other way to get the reasonable initial conditions at

the first time step in the calibration period.

(2) The initial conditions of the hydrological model could be obtained from the processed alternative SM data, which could improve the hydrological efficiency through shortening or skipping the warm-up phase.

(3) LSTM outperforms the traditional curve-fitting method, even using basin-scale to develop the relationship between BTOP model and ERA5-Land SM variables.

Moreover, we suppose the proposed methodology could be applied to any good quality data (e.g., reanalysis, satellite) in temporal and spatial to construct the related initial condition variables in the hydrological model or other models to improve the simulation efficiency. The benefits also cover the data-saving aspect, which is quite precious for the vast poorly- or un-gauged basins worldwide. However, it should be noted that, in this study, only one hydrological model and one global soil moisture dataset were employed to validate the proposed methodology. Future work should address more variables, models,

and datasets to validate its applicability further.

### Data availability

The reanalysis soil moisture data used in this paper are available for download at the following link: https://cds.climate.copernicus.eu/cdsapp#!/dataset/reanalysis-era5-land?tab=form

### Author contributions

Lingxue Liu: Conceptualization, Methodology, Formal analysis, Data Curation, Visualization, Writing- original draft, Writing- review& editing. Tianqi Ao: Methodology, Formal analysis, Funding acquisition, Writing- review& editing, Supervision. Li Zhou: Conceptualization, Methodology, Formal analysis, Data Curation, Funding acquisition, Writing- original draft, Writing- review& editing.



**Competing interests:**

The authors declare that they have no conflict of interest.

**Acknowledgements**

We gratefully acknowledge the Regional Innovation Cooperation Program (2020YFQ0013) and Key R&D Project (2021YFS028) from the Science & Technology Department of Sichuan Province, Key R&D Project (XZ202101ZY0007G) from the Science& Technology Department of Tibet. We thank Climate Data Store (CDS), Copernicus program provides the

ERA5-Land data. We also acknowledge anonymous reviewers for their comments and suggestions that improved this manuscript significantly.

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




**Tables**

**Table 1** The variables of initial condition of BTOP model. The contents in bold are related soil moisture variables.

| Filename | Units | File Descriptions | Default value |
|---|---|---|---|
| *.qi | m³/s | Initial/Input discharge to a grid at one time step | Based on observed |
| *.qo | m³/s | Initial/Output discharge from a grid at one time step | discharge and river route |
| **\*.sdbar** | **m** | **Block average saturation deficit, calculated based on *SD(i)*** | |
| **\*.srz** | **m** | **Root zone storage for the selected time step** | |
| *.sto | m | River channel storage for the Manning's equation routing | |
| **\*.suz** | **m** | **Unsaturated zone storage for the selected time step** | 0 |
| *.svz | m | Vegetation zone storage for the selected time step | |
| *.swz | m | Snow water equivalent for the snow module | |



**Table 2** The features of four hydrological simulation cases.

| Case ID | Source of the SM initial condition | Warm-up | Calibration | Validation | Usage |
|---------|-----------------------------------|---------|-------------|------------|-------|
| Case 1 | Default generated by model | 2002 | | | "Optimal" case |
| Case 2 | Default generated by model | Null | | | Control case: without warm-up |
| Case 3 | ERA5-Land with optimal curve fitting function | Null | 2003-2007 | 2008-2011 | Verifying ERA5-Land could be |
| Case 4 | ERA5-Land with LSTM method | Null | | | used with proper process |



**Table 3** The six curve fitting functions used in this study.

| Curve types | Degree (d) | Functions | Knot number (k) | Restrictions |
|---|---|---|---|---|
| Polynomial | 1, 2, 3, 4 | $C(u) = a_0 u^d + a_1 u^{d-1} + a_2 u^{d-2} + \cdots + a_d u^0$ | / | / |
| Natural Logarithmic | / | $C(u) = a_0 + a_1 \ln a_2 u$ | / | / |
| Natural cubic spline | 3 | $C(u) = \sum_{i=0}^{k} a_i S_{3,i}(u)$ | 3 | $S_{3,i}(t_j) = S_{3,i+1}(t_j) \ S'_{3,i}(t_j) = S'_{3,i+1}(t_j)$ $S''_{3,i}(t_j) = S''_{3,i+1}(t_j)$ $S_3''(a) = S_3''(b) = S_3'''(a) = S_3'''(b)$ |

*Note:* $a_i$ is the constant value; $u$ is the independent variable. For the spline function, consider the interval $[a, b]$ divided by $k$ nodes into $k+1$ pieces, $S_{3,i}(u)$ is the cubic polynomial of the $i$-th piece and $t_j \left(1 \le j \le k\right)$ is the $j$-th knot.






**Table 4** The performance score of curve fitting functions for developing the relationship of SM variables in FRB and SRB. The bold numbers are the highest scores in different basins.

| SM variables | Basins | Polynomial ($d = 1$) | Polynomial ($d = 2$) | Polynomial ($d = 3$) | Polynomial ($d = 4$) | Logarithmic | Natural cubic spline |
|---|---|---|---|---|---|---|---|
| $S_{rz}$ | FRB | 0.188 | 0.641 | 0.404 | 0.501 | 0.333 | **0.739** |
| | SRB | 0.443 | 0.663 | 0.353 | 0.512 | 0.437 | **0.719** |
| $SD$ | FRB | 0.355 | **0.734** | 0.641 | 0.657 | 0.405 | 0.722 |
| | SRB | 0.371 | **0.760** | 0.678 | 0.696 | 0.398 | 0.346 |
| $S_{uz}$ | FRB | **0.741** | 0.639 | 0.510 | 0.363 | 0.499 | 0.610 |
| | SRB | 0.654 | **0.803** | 0.710 | 0.260 | 0.698 | 0.715 |



**Figures**

**Figure 1** The location, DEM, ground observation, and sub-basin of study areas. (a) The Fuji River Basin (FRB); (b) The Shinano River Basin (SRB). Sub-basins are denoted as FRB-1, SRB-1, etc.




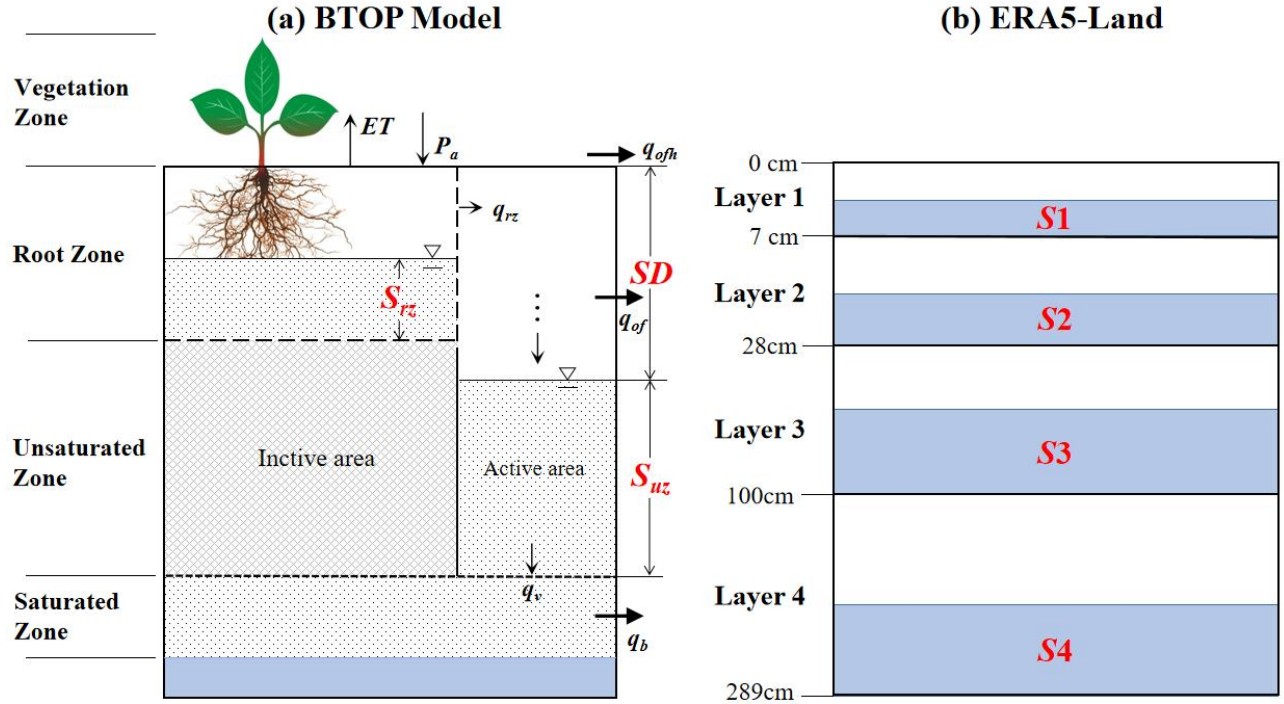

**Figure 2** The structure of soil moisture variables in BTOP and ERA5-Land. (a) The theoretical concept of the BTOP model (modified from Takeuchi et al. (2008), Figure 3); (b) The soil moisture structure (four layers) of ERA5-Land.





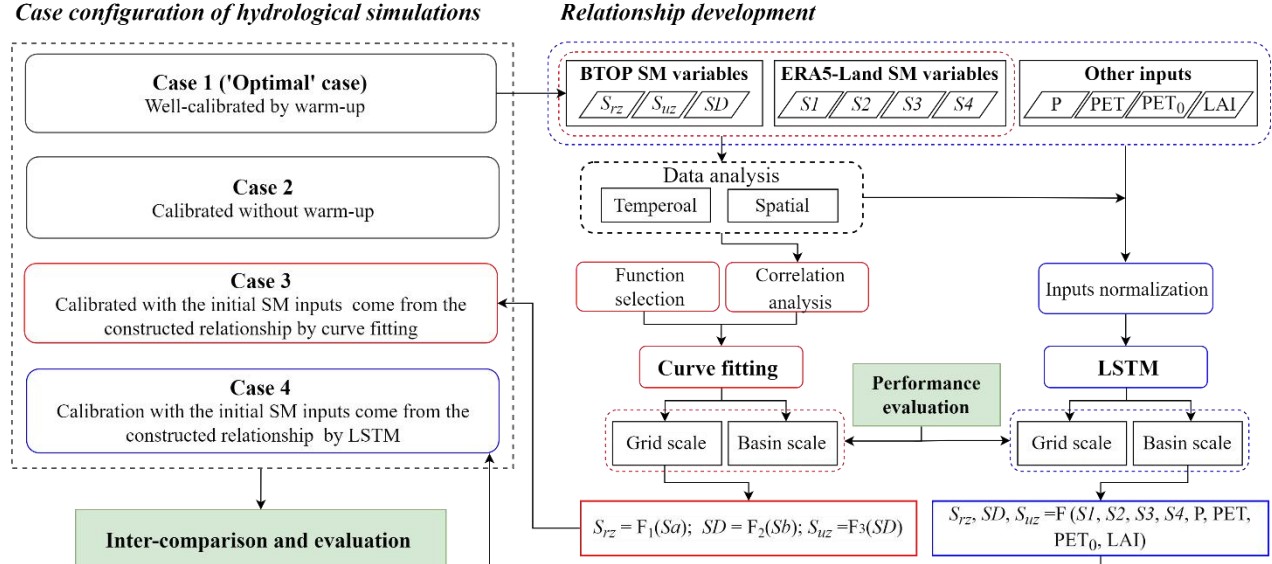

**Figure 3** Framework of this study.


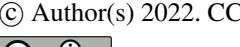



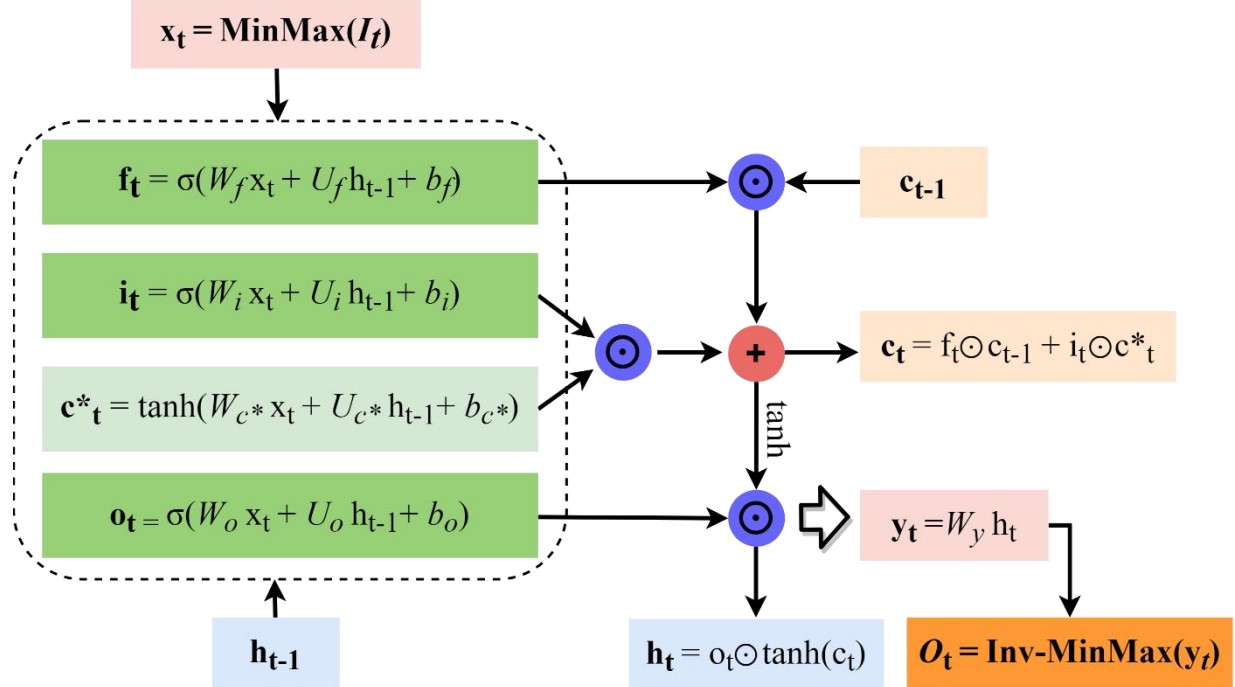

**Figure 4** Schematic diagram of long-short term memory (LSTM). $W$, $U$ and $b$ are the input weights, cyclic weights and bias, respectively. $c_t$, $c*_t$ and $h_t$ are cell state, candidate cell state and hidden state. $i_t$, $f_t$ and $o_t$ are input gate, forget gate and output gate. $x_t$ and $y_t$ are normalized input $I_t$ and output $O_t$. $\oplus$ and $\odot$ are for matrix addition and multiplication respectively.



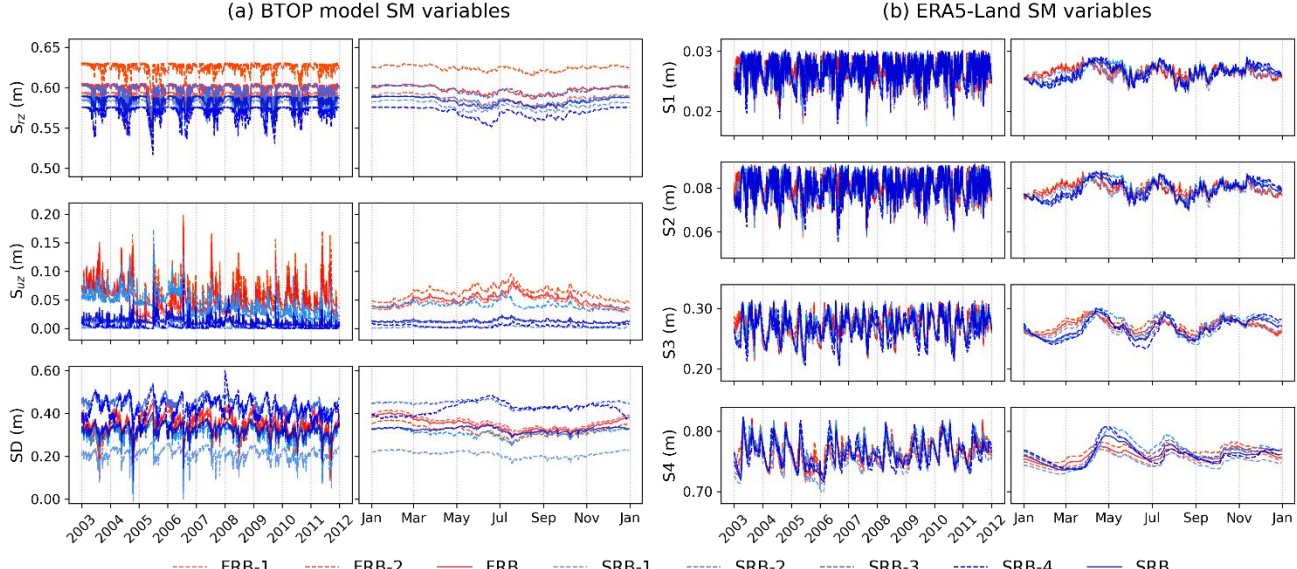

**Figure 5** Temporal performance of BTOP model and ERA5-Land SM variables at daily and annual average daily scale in each (sub-) basins. (a) BTOP model SM variables: $S_{rz}$, $S_{uz}$, and $SD$. (b) ERA5-Land SM variables: $S1$, $S2$, $S3$, and $S4$.



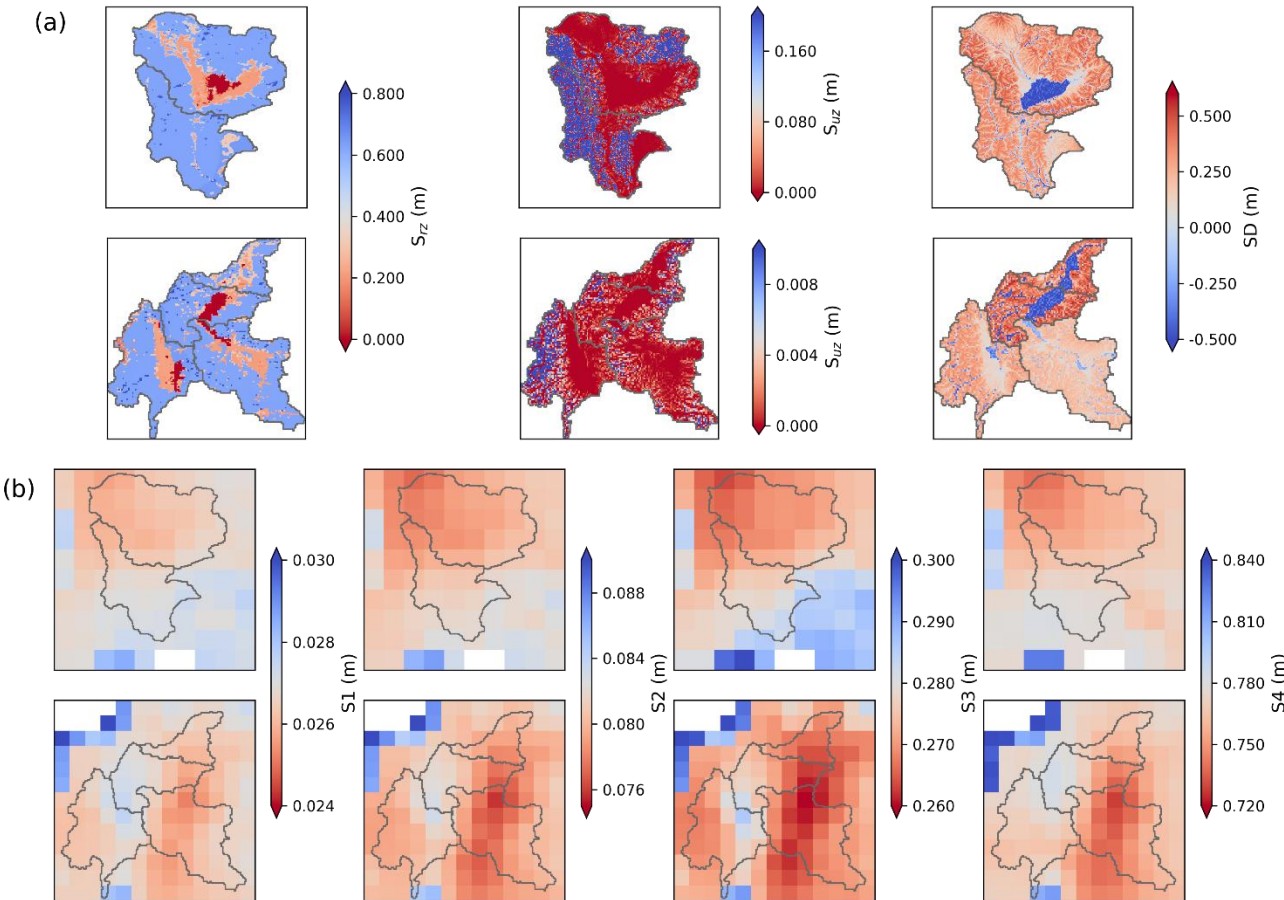

**Figure 6** Spatial performance of the annual average SM variables in the study areas. (a) BTOP model SM variables: *Suz* and *Srz*, *SD*. The resolutions for FRB and SRB are 500 m and 1000 m, respectively. (b) ERA5-Land SM variables: *S1*, *S2*, *S3*, and *S4*. The resolution is 0.1°, approximately 9 km.




**Figure 7** Scatterplots and correlation results of the various normalized SM variable combinations in EXP 1 (i.e., the successive combination of *Srz*, *Suz*, and *SD* with *S1*, *S2*, *S3*, and *S4*) daily and annual average daily scale for FRB and SRB. (a) Daily scale; (b) Annual average daily scale. The values in the figure are processed using Min-max normalization technical.



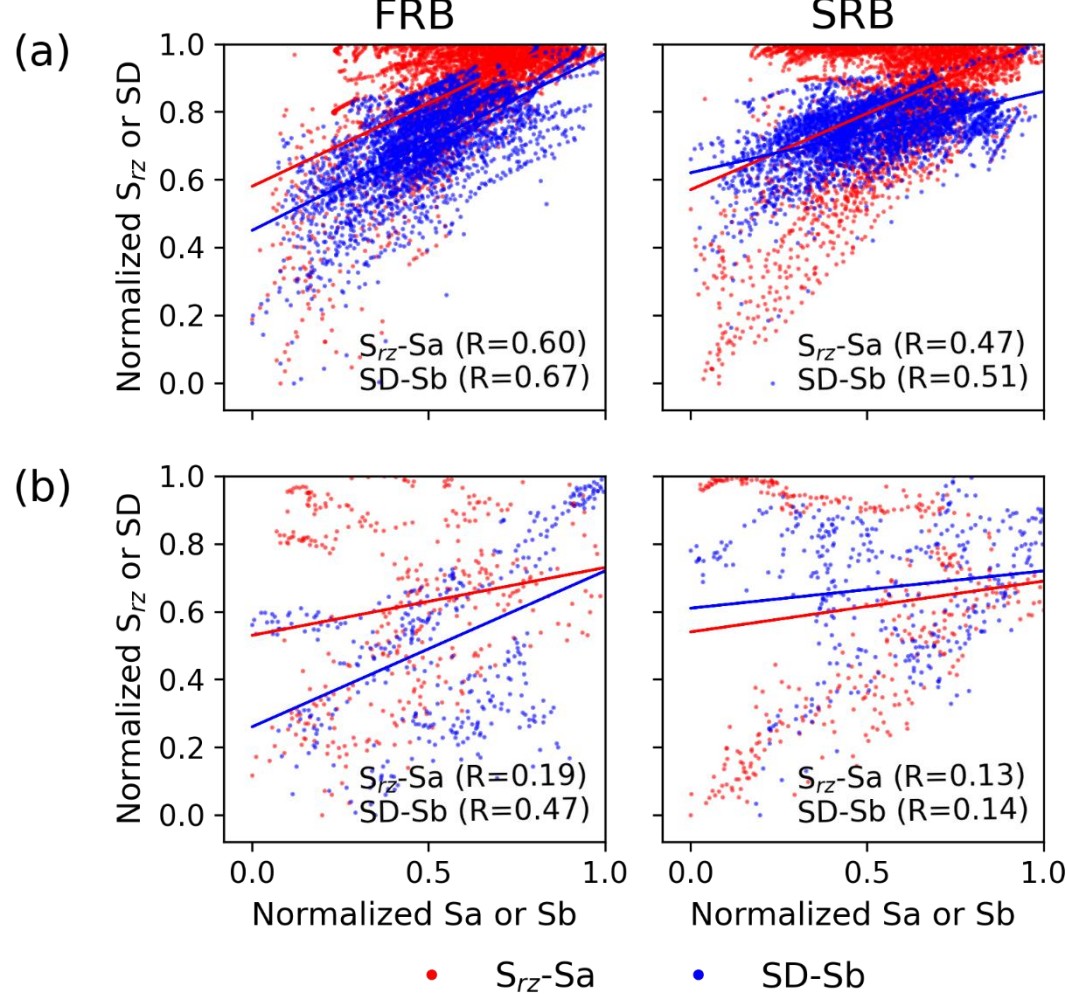

**Figure 8** Scatterplots and correlation results of $S_{rz}$-$Sa$ and $SD$-$Sb$ in EXP 2 in FRB and SRB. (a) and (b) represent the correlations at daily scale and annual average daily scale, respectively. $Sa$ equals the sum of $S1$, $S2$, and $S3$, while $Sb$ equals Depth (289 cm) minus $Sa$. The values in the figure are processed using Min-max normalization technical.





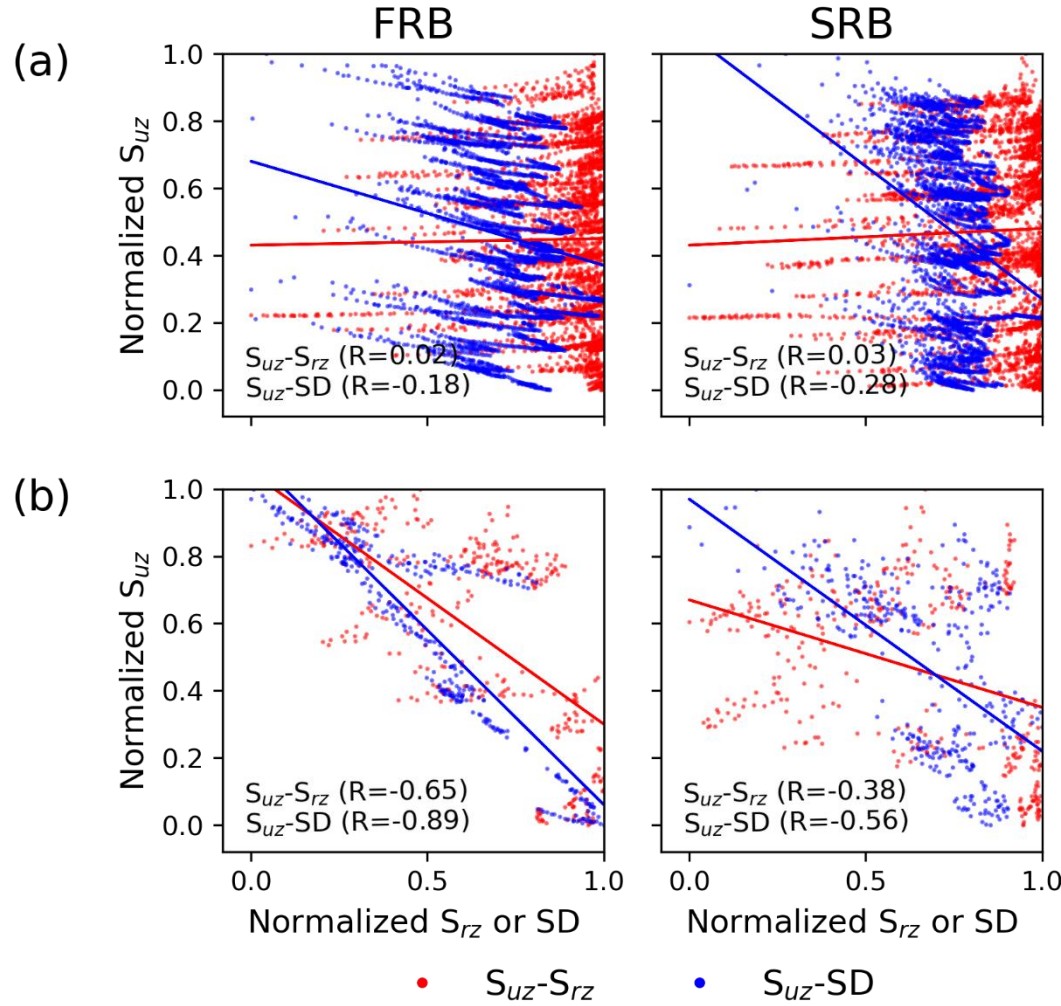

**Figure 9** Scatterplots and correlation results of $S_{uz}$-$S_{rz}$ and $S_{uz}$-$SD$ in EXP 3 in FRB and SRB. (a) and (b) represent daily scale and annual average daily scale, respectively. The values in the figure are processed using Min-max normalization technical.



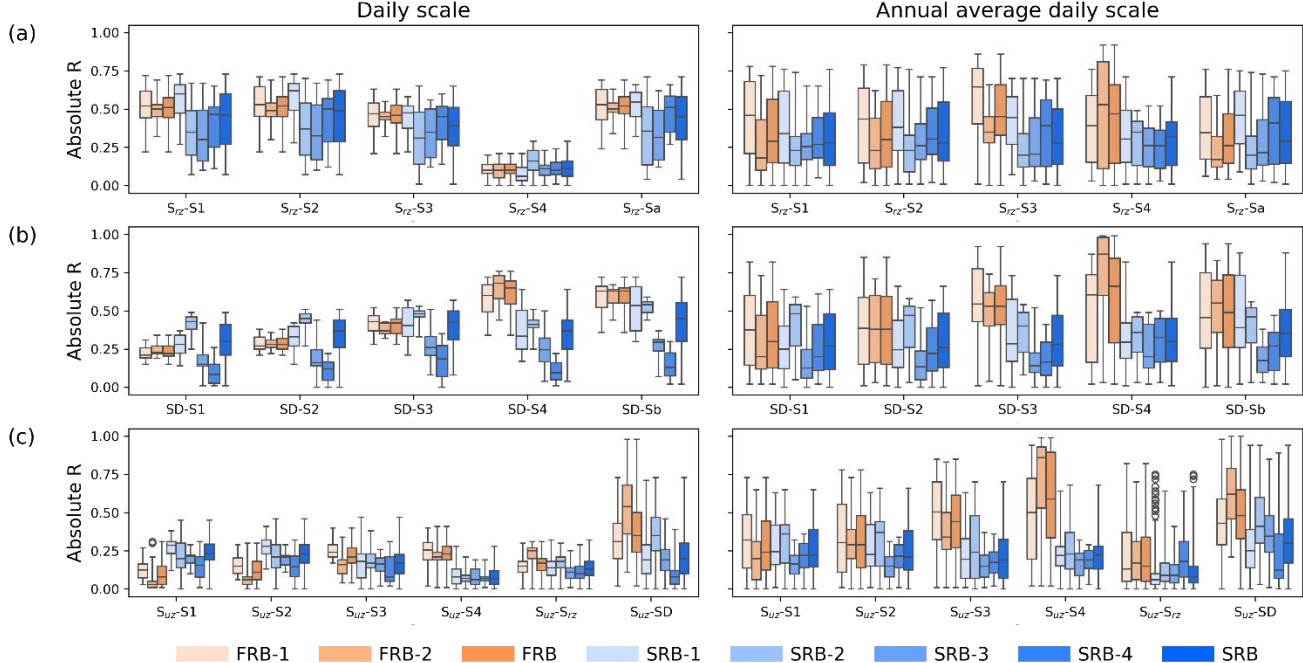

**Figure 10** Boxplots of absolute correlation coefficient values ($R$) between BTOP SM variables ($S_{rz}$, $SD$, and $S_{uz}$) and corresponding SM variables in grid cells at daily and annual average daily scale. (a) $S_{rz}$; (b) $SD$; (c) $S_{uz}$.





**Figure 11** Spatial distribution of the absolute correlation coefficient (*R*) between $S_{rz}$, *SD*, and $S_{uz}$ and corresponding SM variables in the study areas. (a) FRB; (b) SRB. *R* values of $S_{uz}$-$S_{rz}$ and $S_{uz}$-*SD* are obtained from the annual average daily series at BTOP model resolution, while the rest are daily series at 0.1°. $S_{uz}$ has five relationships, and its correlation with *S1* is the worst, which is not shown here for layout purposes.






**Figure 12** Talyor diagram of the corresponding optimal curve fitting functions and LSTM in relationship development at basin- and grid-scales during the training and test period for BTOP SM variables. (a) $S_{rz}$; (b) $SD$; (c) $S_{uz}$. The REF comes from the outputs of Caes 1 that simulated with warm-up. The training periods are shown in red and blue for curve fitting and LSTM, respectively, while orange and purple represent the test periods. The hollow circle denotes the training at basin scale, and its test results are shown by cross markers at grid-scale. Small dots represent the grid-scale results in both training and test periods.







**Figure 13** Performance evaluation of four configured hydrological simulations cases with NSE, KGE' and R. (a) Calibration period; (b) Validation period; (c) Year 2003. The negative values are modified to zero to show the apparent shape of the results. Case 1 is the optimal case with a warm-up. Case 2 is the control case without warm-up. Case 3 and 4 are the cases with initial conditions that come from optimal curve fitting function and LSTM, respectively.