# Peer review of "Technical note: Improving the Initial Conditions of Hydrological Model with Reanalysis Soil Moisture Data"

_EGUsphere, 2022_

## Referee Comment (RC2)

Initial soil moisture condition is one of the most important climatic variables affecting the rainfall-runoff processes. In this study, the author presented a methodology to provide initial soil moisture condition for hydrological model based on the ERA5 Land Surface Reanalysis data. The topic addressed in this study is important and falls within the scope of HESS. However, the experiments were poorly designed on the basis of very strong assumptions that need to be verified. The results presented cannot justify the usefulness of the proposed methodology. With this regard, I suggest to reject this manuscript. Below, my comments on this manuscript are provided.

**Major Comments:**

1. Did the model reach the equilibrium condition with only 1-year warm-up? The author should confirm it.

2. The author should state which parameters were selected for calibration. In addition, based on the description in Section 3.1 and Table 2, the Case 1-4 were calibrated independently, which indicates that the values of parameters used for each case should be different. If that's the case, the performance difference between each case might also be related to the difference in the values of parameters. How does the author consider this? A reasonable configuration for Case 2 would be keeping all other setup the same as Case 1 while initiate the model without warm-up.

3. The design of the methodology builds on three very strong assumptions that need to be justified: 1) line 170, "We consider its simulated variable are the most representative of the hydrological model". How? Based on observation data or model performance? In line 269, the author further stated that "there is no truth value in this study", which is confusing. 2) Line 190, "we assume that its concept is similar to the value of ERA5-Land soil depth (289cm) minus Sa". How? Do the two parameters in the two models represent similar physical process? 3) Line 196, "suppose we could get relationship between on of the BTOP variables and ERA5-Land SM variables". This is a strong assumption. What is it based on?

4. The author should justify the applicability of LSTM for this study. As an advanced data driven method, large amounts of data are needed to use LSTM, which is not the case presented in this study.

5. Instead of only validating the model performance over the period of 2008-2011, which is used for validating the performance of the curve fitting and LSTM, I would suggest the authors further validate the performance of the technique over some independent period.

**Minor Comments:**

1. Title: "Initial Condition" is not accurate, in fact only "Initial Soil Moisture Condition" is discussed in this study

2. Section 2.1, the author should label the places (e.g., Ojiya, Chikuma River Basin, Nagano and Niigata) mentioned in Figure 1. Otherwise, it's hard for the readers to figure out the locations of them. In addition, I would suggest the authors remove the non-related information such as "one of the best rice-producing areas".

3. The manuscript needs revision for language and grammar.

4. In terms of the computational expense, how much the proposed technique is more efficient than the classical warm-up method?

5. Line 125 is confusing to me.

6. Line 144, reference is needed after "hydrological simulation".

7. Line 146, "a better performance" in what?

8. Line 107 uses "①" while Line 186 uses "(1)". Be consistent.

9. Page 10, what is the time interval of simulated discharge.

10. Section 4.1, please consider rephrase the second and third paragraphs, which are really confusing.

---

## Author Comment (AC1)

The topic covered by the Paper is very interesting. The adoption of authoritative (freely available) datasets as support for hydrological analysis or weather-induced impact analysis deserves great attention. In this regard, ERA5 Land represents a valuable example providing data with a very high temporal and spatial resolution (1 hour and 9 km, respectively over the entire Globe).

Thank you for your detailed and informative comments on our work. We appreciate it so much and we'd like to improve our manuscript following your valuable suggestions. Meanwhile, we provide some explanations for your questions and advice.

However, the actual goal of the Article is not clearly identified. The main goal for which hydrological analysis are carried out, has to be identified because it play a key role on the significance (or not) of initial conditions and of the soil moisture data. Furthermore, the two test cases seem to be too large. It entails the adoption of coarse DEM for representing the orography and then the performances could be greatly affected.

This work proposed a way to get the initial conditions of the hydrological model from a global reanalysis dataset. And we conducted the experiments which is one of the applications of the proposed methodology to get the variables for the model initial conditions. Indeed, DEM resolution could affect performance. Actually, we set a hydrological model with a resolution of 500 m and 1 km. However, as the ERA5-Land data has a coarse resolution with 0.1 degrees. So the soil moisture variable from BTOP model were resampled to 0.1 degree to compare with ERA5-Land data.

Furthermore, I suggest introducing, first, the performance of the "best-configuration" hydrological chain. It permits to give confidence on the entire assessment.

Thank you for your suggestion, we will reorganize the structure to better show the results while revising the manuscript.

Discussion and investigation about the reasons entailing the discrepancy between ERA5LAnd and BTOP should be improved: how different are the two soil hydraulic parameter datasets and how the values are computed in the two approaches? The key point is the link among the soil zones of the two approaches: I can understand that the choice is not trivial. Probably an analysis about the water fluxes in ERA5 LAnd could permit clearly linking to the soil profiles in the hydrological model. Furthermore, is it not possible to set the depth of soil profiles in BTOP? However, when you use statistical relationships to "correct" the values, all the physical reasons for which they diverge could play a minor role.

We will improve the discussion and investigation about the reasons entailing the discrepancy between ERA5LAnd and BTOP. As for the question about BTOP soil profile, BTOP model classifies four layers of soil content. Usually, a hydrological model computes these variables by water balance budget. Thus, it is difficult to set depth in the current BTOP model.

Furthermore, some minor suggestions:

The Abstract should be improved. The main topic and the principal Results of the work should be made clearer . A one-sentence about ERA5-Land should complement its introduction. Furthermore, I suggest improving the lexicon (e.g. "luxurious" is not an usual term in scientific literature)

Thank you for your comments. We will improve this part.

General remark: please check the Figures quality. I suppose it should be greatly reduced during the PDF building

Thanks, we will give attention to this point.

L30: to "explore" the uncertainties; it could be better than "minimize"

L31-35: the significance of initial conditions is strictly related to the "memory" of the analysis and then to its duration (as for weather analysis). This aspect should be clarified.

Yes, for weather analysis, like WRF model, the initial conditions play a more

important role than its in the hydrological model.

L43: However, soil moisture represents the key variable as it summarizes the contributions of the different components of the soil water budget (precipitation/infiltration, potential/actual evapotranspiration)

We will describe it in the revised manuscript.

L75: please add information (if available) about the period over which the temperature values have been assessed

Sure, thanks, the temperature had been assessed over 2002-2011 which is consistent with the hydrological simulation period.

§ 2.1 you should try homogenizing the contents in the description of the two Test Cases (e.g. temperature information is missing for the second one)

We will add the information in the revised manuscript.

§ 2.3.1: for long-term analysis, surely, evapotranspiration dynamics should be considered; please provide details and insights :about the choice of using external datasets

As described in Line 133-136, Section 2.3.2, "evaporation module of BTOP model to generate potential interception evaporation (PET0) and potential evapotranspiration (PET)."

L146: ERA5Land is conceptually very far from the other products you have introduced (e.g. satellite data); it should be very important to introduce a paragraph to explain what is a reanalysis is, what is ERA5-Land (e.g. the limitations linked to its horizontal resolution). Furthermore, it should be important to report and compare the soil parameters (e.g. porosity) between the BTOP analysis and ERA5 land. It could significantly influence the results.

We will add related information in the revised manuscript.

L171: please check for typos

We modified it to "They share the exact170 calibration (2003-2007) and validation period (2008-2011), and all cases are auto-calibrated by shuffled complex evolution (SCE-UA) method (Duan et al., 1994) with approximately ten thousand irritations for eight simulations each (four cases for two basins)."

L185: the rationale for the three EXP should be clarified. You are considering a physically-based sub-division with a geometrical one. More details about the coupling are needed.

We explained it in Line 190-199, and we will add more detailed information to support the EXP.

Figure 7: it seems to have a low information content; the scatter plots are quite disperse and then it is hard to identify clear patterns to discuss; furthermore, too many series are retrievable on each plot

Yes, the scatters look disperse for EXP1, that's why we should conduct EXP2 and EXP3.

Figure 13: the investigated variable is not introduced in the graph; please provide additional information

Thank you, we will improve this figure.

Under such premises, in my opinion, the Article is not suitable for publication at this stage but I highly recommend its resubmission after major revisions are implemented

---

## Author Comment (AC2)

Initial soil moisture condition is one of the most important climatic variables affecting the rainfall-runoff processes. In this study, the author presented a methodology to provide initial soil moisture condition for hydrological model based on the ERA5 Land Surface Reanalysis data.

The topic addressed in this study is important and falls within the scope of HESS. However, the experiments were poorly designed on the basis of very strong assumptions that need to be verified. The results presented cannot justify the usefulness of the proposed methodology. With this regard, I suggest to reject this manuscript. Below, my comments on this manuscript are provided.

Thank you for your comments. We appreciate it so much.

We proposed a methodology that well-utilizes the alternative global soil moisture data to improve hydrological simulation efficiency without warm-up by providing the initial conditions of the hydrological model. And the results show the initial conditions of the hydrological model could be obtained from the processed alternative SM data, which could improve the hydrological efficiency through shortening or skipping the warm-up phase.

**Major Comments:**

1. Did the model reach the equilibrium condition with only 1-year warm-up? The author should confirm it.

   Yes, usually, the model could reach equilibrium within one year when conducting daily simulations.

2. The author should state which parameters were selected for calibration. In addition, based on the description in Section 3.1 and Table 2, the Case 1-4 were calibrated independently, which indicates that the values of parameters used for each case should be different. If that's the case, the performance difference between each case might also be related to the difference in the values of parameters. How does the author consider this? A reasonable configuration for Case 2 would be keeping all other setup the same as Case 1while initiate the model without warm-up.

Thank you for your comments. The calibrated parameters in BTOP are shown in Table 1. We will add more information about the BTOP model in the revised manuscript.

Of course, the performance difference between each case might also be related to the difference in the values of parameters. But, our goal is to try to get initial conditions without warmup. As long as the results got improved, the proposed methodology is useful. Moreover, if we use the same parameter set for the simulation, the tuning of the parameters for the different initial conditions is useless.

3. The design of the methodology builds on three very strong assumptions that need to be justified: 1) line 170, "We consider its simulated variable are the most representative of the hydrological model". How? Based on observation data or model performance? In line 269, the author further stated that "there is no truth value in this study", which is confusing. 2) Line 190, "we assume that its concept is similar to the value of ERA5-Land soil depth (289cm) minus Sa". How? Do the two parameters in the two models represent similar physical process? 3) Line 196, "suppose we could get relationship between on of the BTOP variables and ERA5-Land SM variables". This is a strong assumption. What is it based on?

For hydrological simulation, we consider, stimulation using the most data is the best. Our object is to find a way to link BTOP and EAR5-Land. Thus, in Line 267, "It should be noted that, since the BTOP and ERA5-Land variables are come from hydrological and land surface models, and the models' fundament are based on many conception assumptions instead of actual physical law (Liang et al., 1994; Albergel et al., 2012; Frassl et al., 2018; Muñoz Sabater et al., 2021), there is no "truth" value in this study." We provide several references to support our experiment design.

4. The author should justify the applicability of LSTM for this study. As an advanced data driven method, large amounts of data are needed to use LSTM,

which is not the case presented in this study.

Thank you for your comment. We talk about data lacking, which is particularly focused on the discharge data. As for the development of LSTM model, we can put much data like global reanalysis data of other variables like soil moisture, and temperature.

5. Instead of only validating the model performance over the period of 2008-2011, which is used for validating the performance of the curve fitting and LSTM, I would suggest the authors further validate the performance of the technique over some independent period.

Thank you for your comment, we'd like to conduct additional validations for this part.

**Minor Comments:**

1. Title: "Initial Condition" is not accurate, in fact only "Initial Soil Moisture Condition" is discussed in this study

It's our first step to improve the initial conditions of soil moisture. And we will explore other variables in the future.

2. Section 2.1, the author should label the places (e.g., Ojiya, Chikuma River Basin, Nagano and Niigata) mentioned in Figure 1. Otherwise, it's hard for the readers to figure out the locations of them. In addition, I would suggest the authors remove the non-related information such as "one of the best rice-producing areas".

Thank you for your comments. We will improve this part.

3. The manuscript needs revision for language and grammar.

We will carefully improve the language and grammar while revising.

4. In terms of the computational expense, how much the proposed technique is more efficient than the classical warm-up method?

In the aspect of computation expense, they have little difference. The focus of this manuscript is to find a way to skip the warmup phase, therefore, saving discharge data in the ungauged basins.

5. Line 125 is confusing to me.

DEM resolution could affect the performance. Actually, we set a hydrological model with a resolution of 500 m and 1 km. However, as the ERA5-Land data has coarse resolution of 0.1 degrees. So the soil moisture variables from BTOP model were resampled to 0.1 degrees to compare with ERA5-Land data.

6. Line 144, reference is needed after "hydrological simulation".

Thank you for your comments. We will add the reference in the revised manuscript.

7. Line 146, "a better performance" in what?

In the comparison with gauged data, that means EAR5-Land has better consistency with observed soil moisture than other datasets.

8. Line 107 uses "①" while Line 186 uses "(1)". Be consistent.

Thank you. We will modify it.

9. Page 10, what is the time interval of simulated discharge.

We simulated the discharge at a daily time step.

10. Section 4.1, please consider rephrase the second and third paragraphs, which are really confusing.

Thank you. We will improve it.

---

## Author Comment (AC3)

This paper has emphasized the improvement of initial condition of hydrological simulation for many times, but its main work is to make comparison analysis of different given initial conditions. Critically, hydrological simulation without warm-up is misunderstanding. In fact, warm-up and initial conditions given plays different roles in the numerical solution to PDE.

Response:

Thank you for your valuable comments on our work. This work proposed a way to get the initial conditions of the hydrological model from a global reanalysis dataset.

Indeed, we agree with you that the warm-up of the hydrological model is different from the initial conditions given. However, we conducted the experiments which is one of the applications of the proposed methodology to get the variables for the model initial conditions.